# Associations between Dietary Intake of Vitamin D, Sun Exposure, and Generalized Anxiety among College Women

**DOI:** 10.3390/nu14245327

**Published:** 2022-12-15

**Authors:** Fatme Al Anouti, William B. Grant, Justin Thomas, Sharifa AlBlooshi, Spyridon Karras

**Affiliations:** 1Department of Health Sciences, College of Natural and Health Sciences, Zayed University, Abu Dhabi 144534, United Arab Emirates; 2Sunlight, Nutrition and Health Research Center, P.O. Box 641603, San Francisco, CA 94164, USA; 3School of Psychology, Liverpool John Moores University, Liverpool L3 5UX, UK; 4Department of Health Sciences, College of Natural and Health Sciences, Zayed University, Dubai P.O. Box 19282, United Arab Emirates; 5Laboratory of Biological Chemistry, Medical School of Aristotle University, 55535 Thessaloniki, Greece

**Keywords:** anxiety, diet, sun exposure, supplements, United Arab Emirates, vitamin D

## Abstract

Vitamin D insufficiency impacts about half of the population worldwide. Almost one billion individuals across all ages and ethnicities suffer from vitamin D deficiency. Hypovitaminosis D is mainly related to lifestyle choices and habits, such as outdoor activities and food intake. Several studies have demonstrated a correlation between vitamin D status and anxiety symptoms. The main purpose of this study was to investigate the correlation between anxiety and factors including age, vitamin D deficiency, citizens, dietary and supplementary vitamin D intake, along with sun exposure, among a sample of female college students in the United Arab Emirates. A descriptive questionnaire, including a short version of the generalized anxiety disorder scale, food frequency questionnaire, and sun avoidance inventory, was used to assess the relationship between the dietary intake of vitamin D-rich foods and supplements, along with sun avoidance/exposure and generalized anxiety, among a total of 386 female participants aged 18 and above. The findings showed clear evidence that sun avoidance behaviors are strongly associated with an elevated risk of generalized anxiety disorder among adult females in the United Arab Emirates.

## 1. Introduction

Vitamin D deficiency and insufficiency are public health issues globally. Vitamin D levels are especially low in Middle Eastern countries such as the United Arab Emirates (UAE), mostly due to sun avoidance, traditional sartorial style that covers most of the skin, and a lack of food fortified with vitamin D [1,2]. There is a plethora of benefits for adequate vitamin D levels, ranging from calcium metabolism to immune modulation and nervous system health [3].

Insufficient vitamin D levels have been linked to an increased risk of several diseases, such as diabetes, coronary heart disease, respiratory infections, and some cancers [4,5,6]. In addition, vitamin D plays a critical role in brain and nervous system health as it is an essential neuro-steroid hormone involved in brain development, neuroplasticity, and neuroimmunomodulation, and hence may affect mood regulation indirectly [7,8]. Irregularities in brain functions can influence cognitive ability, depression, dementia, autism, and schizophrenia [3]. Furthermore, vitamin D deficiencies have been reported in numerous psychiatric and mental disorders, including anxiety. Inadequate levels were observed in patients with anxiety and depression and were associated with the worsening of symptoms [4,9].

Generalized anxiety disorder (GAD) is one of the most common forms of anxiety disorders. Clinical symptoms may vary for each disorder, but in all cases, anxious physical, cognitive, and behavioral manifestations interfere with a person’s daily life and are difficult to control [10,11]. According to the World Health Organization, as of 2017, 3.6% of the population (approximately 264 million) across the globe suffers from anxiety disorder. Furthermore, females (4.6%) are more affected by anxiety than males (2.6%) [12]. Data about the prevalence of anxiety disorders in the UAE are limited [13]. Moreover, some studies have revealed that several mental health issues might be under- or mis-diagnosed in the UAE, particularly among adolescents [14,15,16]. The most common treatments for anxiety disorders are medication and psychotherapy [10,11]. A few studies have investigated the effect of vitamin D supplementation on individuals with GAD affected by vitamin D insufficiency, with conflicting results [17,18]. One study reported that supplementation was effective in increasing the levels of serotonin and improving mood [17]. In contrast, a study on healthy adult women reported the lack of association between vitamin D supplementation and improved mood-related outcomes, including anxiety and depression [18]. Despite the research debate, the potential of vitamin D for improving mental health remains promising; however, its full effect on mood modulation is yet to be fully elucidated. The fundamental purpose of this study is to investigate the correlation between anxiety and factors including age, vitamin D deficiency, dietary and supplementary vitamin D intake, along with sun exposure, among a sample of female college students in the UAE.

## 2. Materials and Methods

### 2.1. Study Design

This exploratory study utilized a cross-sectional, correlational design and was conducted in the capital city of the UAE, Abu Dhabi, between March 2021 and April 2021. Based on the revision of relevant studies about dietary vitamin D intake, sun exposure, and anxiety, a compiled questionnaire was developed from a previously validated questionnaire, which included a food frequency questionnaire (FFQ) and a sun avoidance inventory (SAI) [19] in addition to GAD-7 [20].

Dietary data were collected using a modified FFQ, a self-administered instrument that considers the self-reported recall of the intake of vitamin D-rich foods (naturally and fortified) for the last 4-week period. The modified FFQ was based on previous studies that included local food items [19,21]. The cut-off for adequate intake of vitamin D-rich foods was 19 and thus FFQ scores ranging from 20 to 45 reflected sufficient intake. The sun avoidance inventory (SAI) assesses sun avoidance attitudes and behaviors across six factors: cosmetic/aesthetic; health and safety; transport; occupational; recreational; and sartorial. Sun exposure was evaluated using a modified version of the SAI, which is a questionnaire designed to assess attitudes towards sun avoidance. The SAI uses a five-point Likert scale to record a participant’s responses, from strongly agree to strongly disagree, graded 0 to 4, depending on the item direction. A high overall score on the SAI indicates that the individual minimizes sun exposure, while a low score reveals that the individual maximizes exposure to sun [22]. The GAD-7 questionnaire was used to determine the severity of anxiety. The GAD-7 score is calculated by assigning scores of 0–3 to the response categories: “not at all”; “several days”; “more than half the days”; and “nearly every day”. The GAD-7 total score for the seven items ranges from 0 to 21, with 0–4 indicating minimal anxiety, 5–9 mild anxiety, 10–14 moderate anxiety, and 15–21 severe anxiety [20].

Participants were also asked to report if they were diagnosed within the span of the past 12 weeks with vitamin D deficiency and whether they were taking any vitamin D supplements.

### 2.2. Study Participants and Data Collection

Information about the study and its objectives, along with the questionnaire, was distributed electronically with explicit inclusion and exclusion criteria and a consent form. The questionnaire was advertised by both random and snowball sampling through social media to two main universities in Abu Dhabi. A total of 425 subjects participated in the survey, but only 386 were considered for the final analysis after the exclusion of surveys with incomplete data. The inclusion criteria were as follows: being healthy; residing in the UAE; aged 18 years and above; and attending university in Abu Dhabi. Any participant that did not meet the criteria was excluded from the study.

### 2.3. Data Analysis

The data collected were analyzed by SPSS version 27.0 (IBM, Armonk, NY, USA). The association of the main variable anxiety with age, vitamin D deficiency, citizenship, dietary and supplementary vitamin D intake, along with sun exposure, was examined. In the descriptive analysis, demographic and main variables were summarized using frequency distributions. To explore the association between general anxiety and vitamin D-related variables, we conducted a bivariate correlational analysis. Moreover, to assess vitamin D-related variables as predictors of anxiety status, we performed multivariate logistic regression analyses with anxiety symptom status according to the GAD-7 cut-off as the dependent variables.

## 3. Results

### 3.1. Sample Characteristics and Descriptive Statistics of Key Variables for Participants

We performed a descriptive analysis to obtain sample characteristics. The measures of psychopathology (GAD-7 data) were right-skewed but all other continuous variables (age, FFQ, and SAI score) were normally distributed. The GAD-7 has a well-established screening cut-off, with scores ≥ 10 considered indicative of moderate levels of clinically significant symptomatology. Table 1 shows the sample characteristics, including the percentages of participants scoring above the screening cut-offs on the GAD-7. The majority of participants (approximately 88%) were 25 years old or younger and 65% reported being vitamin D-deficient. In addition, those taking vitamin D supplements (range of 1000–50,000 IU weekly or 1000 IU daily) accounted for around 58% (Figure 1). Emirati nationals accounted for 74% of the survey participants, and 67% of the sample’s GAD-7 scores lay above cut-off, suggesting significantly high levels of anxiety amongst the participants (Table 1).

The descriptive statistics for the study’s continuous variables are detailed in Table 2. Notably, the mean GAD-7 and SAI scores were 12 (SD = 6) and 17 (SD = 5), respectively, suggesting relatively high levels of anxiety and avoidance of sun across the whole population. On the other hand, the mean FFQ score of 10 (SD = 8) revealed an inadequate dietary intake of vitamin D-rich foods (Table 2).

### 3.2. Correlational Analysis between Key Study Variables for All Participants

To explore the association between general anxiety and vitamin D-related variables, we undertook a bivariate correlational analysis. We initially used scatter plots to explore the key variables to rule out potential nonlinear relationships. The data indicated the hypothesized relationships between anxiety and three of the vitamin D-associated variables. Table 3 details the two-tailed correlation coefficients between the relevant variables.

### 3.3. Bivariate and Multivariate Logistic Regression Analysis

In line with the aims of the study and to further assess vitamin D-related variables as predictors of anxiety status, we conducted multivariate logistic regression analyses with anxiety symptom status (above or below the GAD-7 cut-off) as the dependent variables. The predictor variables were age, VTD status, SAI, and FFQ; nationality and supplement use were not included in the multivariate analysis as the bivariate analysis revealed these two variables were not predictive of anxiety status. All the variables were retained as predictive of anxiety symptom status in the multivariate analysis. In short, younger participants who reported a history of VTD deficiency, a less VTD-rich diet, and greater sun avoidance were at higher risk of clinically significant anxiety symptoms. Table 4 details the results of this analysis.

### 3.4. Vitamin D Supplement Use by Participants

Among those who were taking vitamin D supplements, 27% reported taking weekly doses of 10,000 IU. Almost 16% consumed a daily dose of 1000 IU and around 9% took the monthly dose of 50,000 IU, while 48% did not take any supplements (Figure 1). Furthermore, the GAD-7 was included in the questionnaire given to these participants and according to the results shown in Table 1, 67% of the sample’s scores lay above the cut-off, suggesting significantly high levels of anxiety amongst them. In Table 4, however, the measure of relative risk, otherwise called odds ratios, is obtained. The relative risk in this case is the risk of scoring above the anxiety cut-off. Thus, those who are vitamin D-deficient are more likely to have elevated anxiety levels (OR = 1.52). Moreover, those with high FFQ scores were less likely to score above the anxiety cut-off (OR = 0.97). Age was also correlated and being younger meant the risk was higher (OR = 0.93).

## 4. Discussion

The results of this research demonstrated interesting conclusions regarding vitamin D dietary intake, sun avoidance, and anxiety among a sample of adult females from universities in Abu Dhabi, the capital of the UAE. A high rate (65%) of VTD deficiency was reported by the female subjects. Furthermore, the present study confirmed a clear pattern of sun avoidance, as is reflected by the high SAI mean score, and an inadequate intake of vitamin D-rich foods, as is shown by the low FFQ mean score (Table 2). These data are concordant with previous research findings among female university students in the UAE, whose limited dietary vitamin D intake and minimal sun exposure deterred them from attaining adequate vitamin D status [19,23].

The correlation analysis and multivariate logistic regression analysis to extrapolate associations with GAD demonstrated a significant correlation between anxiety symptoms and decreasing levels of dietary vitamin D intake and sun exposure. This finding is in line with studies from other populations that also reported a similar relationship [24,25]. According to our results, higher SAI and lower FFQ scores were strongly correlated with reported vitamin D deficiency and a greater risk for GAD. Moreover, a history of vitamin D deficiency was found to be associated with greater risks of GAD. The data are in concordance with previous findings from relevant research which highlighted the correlation between vitamin D deficiency and anxiety, where patients with vitamin D deficiency had higher Hospital Anxiety and Depression Scores (HADS) compared to patients with insufficient or normal levels [26]. Another study conducted on the Saudi population found a similar positive correlation between vitamin D deficiency and anxiety [24]. However, other researchers have reported a lack of association between vitamin D status and anxiety. One study measured 25(OH)D serum levels among young Australian women and used mental health measures to investigate whether a link could be established. The results concluded the absence of an association between 25(OH)D serum levels and mental health status, including anxiety [27]. However, these results could be explained by the low prevalence of severe vitamin D deficiency and mental health symptoms in the examined sample, unlike with other studies. In addition, sunlight exposure prompts the production of dermal nitric oxide and the release of serotonin and melatonin hormones in the brain. Nitric oxide helps to diminish anxiety and depression by alleviating inflammation and oxidative stress, as well as amplifying the production of serotonin and melatonin which are associated with sleep regulation, improving mood, and reducing anxiety; thus, the important non-vitamin D-relevant benefits for mental wellbeing by the sun cannot be ignored [28].

Higher measures of sun avoidance habits, as depicted by SAI scores among the female college students examined in our study, correlated with a greater risk of GAD. Moreover, SAI was a predictor of GAD status. Sun avoidance habits by students in the UAE can be attributed to the generally hot and humid weather in the country for most of the year, particularly in the summer months. Therefore, outdoor activities are avoided, limiting exposure to the sun. Furthermore, Emirati females dress in clothing that covers most of the body, except for the hands and face, for cultural and religious reasons. The choice of clothing can hence additionally contribute to reduced sun exposure and consequentially lower vitamin D levels [29]. A previous study investigated the relationship between vitamin D deficiency and depression among UAE college students and found a significant link due to sun avoidance habits [19]. Students with low vitamin D dietary intake (FFQ) were also at a greater risk of depression. Moreover, the difference in sartorial habits between Emiratis and non-Emiratis in our study corroborates the higher prevalence of vitamin D deficiency among Emiratis because most of the body is concealed.

Vitamin D intake through diet is relatively difficult given the few sources available. Additionally, vitamin D sources, such as fish and mushrooms, can have variable amounts of vitamin D, further complicating the intake of sufficient amounts from food [30]. Few studies have demonstrated that vitamin D supplementation can alleviate symptoms of GAD [8,9,17]. In one study, researchers randomized participants with low 25(OH)D levels into control or daily vitamin D supplementation groups and followed up for 6 months. Using the Hamilton Depression Rating Scale-17, Revised Social Anhedonia Scale, Revised Physical Anhedonia Scale, and the Hamilton Anxiety Rating Scale-14, the researchers found that, while there were no significant differences in depression symptoms among the vitamin D supplementation group, there was a significant improvement in anxiety symptoms [8].

Our data illustrated that in addition to SAI and FFQ, age was also correlated with risk of GAD, where a negative correlation was found with GAD. Previous findings by other researchers had highlighted that certain environmental and cultural risk factors for anxiety and mental disorders are common in the UAE and other Arab countries. These include large family units, a non-parental caregiver, and consanguineous marriages, which mostly affect younger people [13,16,31,32]. A few studies among UAE college students revealed the high prevalence of body dissatisfaction [33,34]. Body obsession, as well as family and societal factors, plays a significant role in the mental wellbeing of young people living in the UAE [35].

## 5. Strengths and Limitations

This study examined the associations of predictors of vitamin D deficiency among a vulnerable subpopulation of young adult females in the UAE. Low vitamin D status and anxiety among this subpopulation could signify health deterioration and the need for action. Our findings have valuable public health implications regarding the physical wellbeing of young adult females and could inform future larger screening and intervention-based studies. Moreover, the study included both Emiratis and non-Emiratis to give a better representation of the UAE population.

Despite the strength of the study, several limitations are to be acknowledged, including the cross-sectional nature of design, which does not confirm causation. Thus, the findings remain as correlational and require further validation. Moreover, the generalizability of the results is restricted because participants were from one sex with a confined age range. Additional longitudinal studies, causality models, and consideration of other settings where sunlight exposure is not limited are needed for further cross-validation. In this study, we used vitamin D dietary and supplement intake and sun avoidance as determinants to establish vitamin D status instead of serum 25(OH)D levels, which are frequently used as clinical indicators for the assessment of vitamin D status. In this research, clinical assessment was not performed and there was also no adjustment for season; however, the research was conducted between March and April, which marks the end of the cool winter season in the UAE. Moreover, previous investigations regarding seasonal variation in the UAE had revealed that serum 25(OH) levels are highest in April and lowest in October, which marks the end of the hot summer season [19].

Lastly, genetic factors cannot be ignored due to the role of genetic variants within vitamin D metabolism in affecting vitamin D status. Nonetheless, this study provides valuable data regarding mental wellbeing, specifically anxiety, and the predictors of vitamin D status among young adult females residing in the UAE.

## 6. Conclusions

In conclusion, this study used vitamin D dietary and supplement intake and sun avoidance as determinants to establish vitamin D status among a vulnerable sample of young adult females from universities in Abu Dhabi, the capital of the UAE, and examined associations with generalized anxiety. The findings demonstrated a prominent association between sun avoidance and a low intake of vitamin D-rich foods and supplements on the one hand and anxiety on the other. This suggests that rectifying vitamin D levels may be a convenient, cost-effective, and low-risk method to improve anxiety and mental health status in general. However, any conclusive results regarding the potential of vitamin D supplements for relieving symptoms of anxiety, more likely at the neurochemical level, should be interpreted with caution before compelling evidence is provided by additional RCTs and MR studies.

## Figures and Tables

**Figure 1 nutrients-14-05327-f001:**
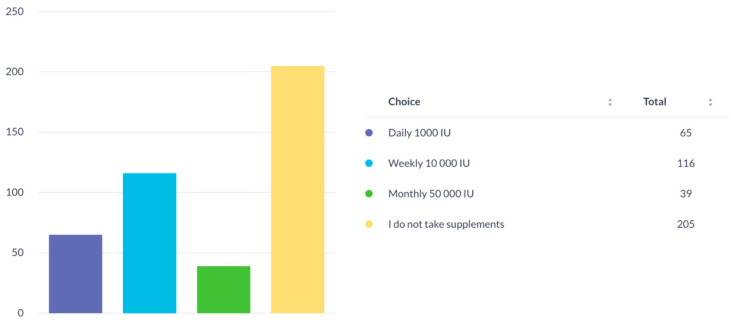
Vitamin D supplement use by participants.

**Table 1 nutrients-14-05327-t001:** Sample characteristics of the study participants.

Variable	Frequency (%)
Citizenship	
Emirati	286 (74%)
Non-Emirati	100 (26%)
Age Group	
Above 25 years	48 (12%)
25 years or below	338 (88%)
Diagnosed with VTD deficiency	
Yes	250 (65%)
No	136 (35%)
VTD supplementation	
Yes	223 (58%)
No	163 (42%)
GAD-7—Anxiety Above Cut-off	
Yes	259 (67%)
No	127 (33%)

**Table 2 nutrients-14-05327-t002:** Descriptive statistics for key variables for study participants.

	Age	GAD-7	FFQ	SAI
Median	21	13	8	17
IQR	18–23	8–17	4–15	13–20
Mean	22	12	10	17
SD	6	6	8	5
Min	18	0	0	7
Max	25	21	40	35

GAD-7 = Generalized Anxiety Disorder; SAI = sun avoidance inventory; FFQ = food frequency questionnaire.

**Table 3 nutrients-14-05327-t003:** Correlation analysis between key study variables for all participants.

	GAD-7	SAI	FFQ	VTD	Suppl.	Citizen
Age	−0.25 ***	−0.12 *	0.02	0.08	0.15 **	0.04
GAD-7		0.13 **	−0.12 *	0.10 *	0.03	−0.06
SAI			0.10 *	−0.10	−0.03	−0.06
FFQ				−0.10 *	0.09	0.08
VTD					0.20 ***	0.12 *
Suppl.						0.01

GAD-7 = Generalized Anxiety Disorder; SAI = sun avoidance; FFQ = food frequency; VTD = history of vitamin D deficiency; Suppl. = currently taking vitamin D supplement (Yes = 1, No = 0); Citizen = UAE citizen (Yes = 1, No = 0). * *p* < 0.05, ** *p* < 0.01, *** *p* < 0.001.

**Table 4 nutrients-14-05327-t004:** Bivariate (OR) and multivariate (AOR) logistic regression predicting clinically significant generalized anxiety symptom scores.

	*N*	Above Anxiety Cut-Off	Odds Ratio	Adjusted Odds Ratio
VTD (Yes)	250	176 (70.4%)	1.52 (1.01–2.36) *	1.65 (1.04–2.61) *
FFQ			0.97 (0.94–1.00) *	0.97 (0.93–0.99) *
SAI			1.05 (1.01–1.10) *	1.05 (1.00–1.10) *
Age			0.93 (0.90–1.00) ***	0.93 (0.90–1.00) ***

Notes: *N* = 386; VTD = history of vitamin D deficiency; FFQ = food frequency; SAI = sun avoidance inventory * *p* < 0.01, *** *p* < 0.001.

## Data Availability

The data presented in this study are available on request from the corresponding author.

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
