# Peer review of "Associations between Dietary Intake of Vitamin D, Sun Exposure, and Generalized Anxiety among College Women"

_nutrients, 2022, doi:10.3390/nu14245327_

Round 1

Reviewer 1 Report

A major concern

The authors did not consistently focus on a clear aim of this study. The manuscript is hard to follow up, particularly methods and results.

Line 21 In this study, the relationship between anxiety and determinants of vitamin D status including vitamin D dietary and supplement intake along with sun exposure is investigated specifically among adult female students attending University in the United Arab Emirates, a country that is blessed with sun almost all year round.

Line 109 - The three main outcome variables were anxiety, sun exposure and vitamin D intake from food and supplements.

For me, the following seems a more appropriate description for the purpose of this study.

The main purpose of this study was to investigate the correlation between anxiety and factors including age, vitamin D deficiency, citizen, dietary and supplementary vitamin D intake along with sun exposure among 386 college females in the UAE.

Abstract:

Hypovitaminosis is mainly related to lifestyle choices and habits such as outdoor activities and food intake.

Please add “D” in Line 18.

Several studies demonstrated a correlation between vitamin D and reduction of depression and anxiety symptoms.

Please change into “Several studies demonstrated a correlation between vitamin D status and anxiety symptoms.”

Line 22

Please change “is” into “was”.

Introduction

The second paragraph and the third paragraph should be rewritten.

Author Response

We thank the reviewer for the very valuable comments and suggestions. We have addressed all the mentioned points and believe the manuscript has now improved significantly. Below are the detailed point by point responses.

A major concern

The authors did not consistently focus on a clear aim of this study. The manuscript is hard to follow up, particularly methods and results.

 We thank the reviewer for such valuable comments, we had rewritten the last 2 paragraphs of the introduction to provide a clear background and focused objectives for the methodology and results accordingly.

Line 21 In this study, the relationship between anxiety and determinants of vitamin D status including vitamin D dietary and supplement intake along with sun exposure is investigated specifically among adult female students attending University in the United Arab Emirates, a country that is blessed with sun almost all year round.

 Line 109 - The three main outcome variables were anxiety, sun exposure and vitamin D intake from food and supplements.

 For me, the following seems a more appropriate description for the purpose of this study.

The main purpose of this study was to investigate the correlation between anxiety and factors including age, vitamin D deficiency, citizen, dietary and supplementary vitamin D intake along with sun exposure among 386 college females in the UAE.

Response:

Thank you for the valuable comment, we have adjusted the objective in the introduction and in the methods section as follows:

Introduction: The main purpose of this study was to investigate the correlation between anxiety and factors including age, vitamin D deficiency, citizen, dietary and supplementary vitamin D intake along with sun exposure among a sample of college females in the UAE. (Line 20)

The association of the main variable anxiety with age, vitamin D deficiency, citizenship, dietary and supplementary vitamin D intake along with sun exposure was examined. (Line 112)

Abstract:

Hypovitaminosis is mainly related to lifestyle choices and habits such as outdoor activities and food intake.

Please add “D” in Line 18.

 Thank you for this comment, the word had been edited.

Several studies demonstrated a correlation between vitamin D and reduction of depression and anxiety symptoms.

Please change into “Several studies demonstrated a correlation between vitamin D status and anxiety symptoms.”

 Thank you for this comment, we have changed the sentence accordingly.

Line 22

Please change “is” into “was”.

Response: We have changed the whole sentence now which reads as follows: “The main purpose of this study was to investigate the correlation between anxiety and factors including age, vitamin D deficiency, citizen, dietary and supplementary vitamin D intake along with sun exposure among a sample of college females in the United Arab Emirates.”

 Introduction

The second paragraph and the third paragraph should be rewritten.

We thank the reviewer for such valuable comments about this section, we had rewritten the last 2 paragraphs of the introduction to provide a clear background and focused objectives.

Reviewer 2 Report

Review of the article „Associations between Sun Exposure, Dietary intake of Vitamin D rich Foods and Supplements and Anxiety among College Female Students“

I read a paper that examined  the correlation between dietary and supplemental intake of vitamin D with sun exposure and general anxiety disorder among a sample of female students aged years

UAE. The studied topic is very interesting and current and I definitely support research on this topic because Vitamin D deficiency is public health issue globally. In the paper, it would be good to show the difference between Emiratis and non-Emiratis because of the dress in clothing that covers most of the body, except for the hands and face for cultural and religious reasons.

Specific corrections and suggestions are listed below. 

Introduction:

Line 60: when a few authors are mentioned, then several references must be given.

Materials and Methods:

Line 82: the reference given (21) for the modified FFQ does not fit

Results:

In Table 1. „Sample Characteristics, Demographics, and Categorical Variables“ the division of participants by age group is not clear. That division needs to be better explained.

Discussion:

Line 231: when „Multiple studies are mentioned“, then several references must be given

Author Response

We thank the reviewer for the insightful comments and reiteration of the importance of addressing public health concerns about links between Vitamin D deficiency and anxiety among vulnerable subpopulations.

We have addressed all the mentioned points and believe the manuscript has now improved significantly. Below are the detailed point by point responses.

Review of the article „Associations between Sun Exposure, Dietary intake of Vitamin D rich Foods and Supplements and Anxiety among College Female Students“

I read a paper that examined the correlation between dietary and supplemental intake of vitamin D with sun exposure and general anxiety disorder among a sample of female students aged years

UAE. The studied topic is very interesting and current and I definitely support research on this topic because Vitamin D deficiency is public health issue globally. In the paper, it would be good to show the difference between Emiratis and non-Emiratis because of the dress in clothing that covers most of the body, except for the hands and face for cultural and religious reasons.

We thank the reviewer for his insightful comment to improve the quality of the manuscript. We had elaborated more about the difference in sartorial factors for Emiratis versus non-Emiratis. We have addressed also the specific comments listed below. We have added on line 232 the following: “Moreover, the difference in sartorial habits between Emiratis and non-Emiratis in our study corroborates the higher prevalence of vitamin D deficiency among Emiratis because most of the body is concealed”. 

Specific corrections and suggestions are listed below. 

Introduction:

Line 60: when a few authors are mentioned, then several references must be given.

Thank you for this note. We had modified the sentence to read as follows: “Few researchers had investigated the effect of vitamin D supplementation on individuals with GAD affected by vitamin D insufficiency and had demonstrated conflicting results [17,18]”.

Materials and Methods:

Line 82: the reference given (21) for the modified FFQ does not fit

Thank you for this note, we have added reference 19 to cite the modified FFQ, but reference 21 to show the consideration of local food for the FFQ.

Results:

In Table 1. „Sample Characteristics, Demographics, and Categorical Variables”, the division of participants by age group is not clear. That division needs to be better explained.

Thank you for this important observation, we had adjusted the cut off to 25 years or younger and above 25 years in the table and text to avoid confusion.

Discussion:

Line 231: when „Multiple studies are mentioned“, then several references must be given.

Thank you for this remark, we have adjusted the Multiple studies to few studies and added references, 8,9,17.

Round 2

Reviewer 1 Report

There were three major concerns.

First, Figure 1 is not included in the manuscript.

Second, their description regarding Table 2 was not matched with Table 2.

Such as, the mean FFQ score of 8 (SD=7.75) revealed an inadeqaute dietary intake of vitamin D rich foods.

What was the cutoff indicating adequate dietary intake of vitamin D rich foods?

Third, the statements of the results were incorrect. The findings in Table 4 revealed only two variables (i.e.: a diagnosis of vitamin D deficiency within the past 12 weeks and food frequency questionnaire) as significant predictors to predict generalized anxiety symptom scores clinically because adjusted odds ratios of SAI (Sun avoidance inventory) and age were involved with 1.0. Therefore, the title was not matched with the findings of the study.

Some minor concerns:

1.      The following subtitles were not matched with the text contents.

3.1. Vitamin D Determinants and Generalized Anxiety

3.2. Correlational Analysis

3.3. Multivariate Logistic Regression Analysis

3.4. Data about the Use of Vitamin D supplements

2.      The following titles should be rewritten.

Table 1. Sample Characteristics, Demographics, and Categorical Variables.

Table 2 Descriptive Statistics for Outcome Variables

3.      What did FFR stand for in Table 3?

4.      There were some wrong spellings.

Author Response

We greatly thank reviewer 1 for the thorough revision and the very valuable comments. We have addressed all comments and now feel the manuscript is of better quality thanks to these remarks and suggestions

Comments by Reviewer 1

Comments and Suggestions for Authors

There were three major concerns.

First, Figure 1 is not included in the manuscript

We added Figure 1 in a separate file as per Nutrients’ format, we apologize if the second version did not imply this. We have attached the figure again. We also added more information about those not taking supplements so that the paragraph now reads as follows:

“Among those who were taking vitamin D supplements, 27% reported taking weekly doses of 10,000 IU. Almost 16% consumed a daily dose of 1000 IU and around 9% took the monthly dose of 50,000 IU; while 48% did not take any supplements (Figure 1)”.

Second, their description regarding Table 2 was not matched with Table 2.

Such as, the mean FFQ score of 8 (SD=7.75) revealed an inadeqaute dietary intake of vitamin D rich foods.

We sincerely apologize for this error, we had adjusted the text to match all values in Table 2 now since we rounded significant figures.

What was the cutoff indicating adequate dietary intake of vitamin D rich foods?

The cutoff for FFQ to indicate adequate intake of vitamin D rich foods was 19; we have clarified this information in the methods section line 89

The cutoff for adequate intake of vitamin D-rich foods was 19 and thus FFQ scores ranging from 20 to 45 reflected sufficient intake

 Third, the statements of the results were incorrect. The findings in Table 4 revealed only two variables (i.e.: a diagnosis of vitamin D deficiency within the past 12 weeks and food frequency questionnaire) as significant predictors to predict generalized anxiety symptom scores clinically because adjusted odds ratios of SAI (Sun avoidance inventory) and age were involved with 1.0. Therefore, the title was not matched with the findings of the study.

 Thank you for this remark. For logistic regression, the continuous variable (age) was used as predictors, only the dependent variable (anxiety) was dichotomous (1 or 0). While continuous variables can not be used as output variables in logistic regression, the (continuous variables) can be used as predictors and do not need to be dichotomized (1 and 0). For example, there must be two or more independent variables, or predictors, for a logistic regression. The IVs, or predictors, can be continuous (interval/ratio) or categorical (ordinal/nominal).

Cleophas, T.J., Zwinderman, A.H. (2016). Logistic Regression with a Continuous Predictor (55 Patients). In: SPSS for Starters and 2nd Levelers. Springer, Cham. https://doi.org/10.1007/978-3-319-20600-4_37

Some minor concerns:

  1. The following subtitles were not matched with the text contents.

We have modified the titles as follows

3.1. Vitamin D Determinants and Generalized Anxiety 

Sample Characteristics and Descriptive Statistics of Key Variables for Participants

3.2. Correlational Analysis

We modified to: Correlationl Analysis between Key Study Variables for All Participants

3.3. Multivariate Logistic Regression Analysis

Changed to Bivariate and Multivariate Logistic Regression Analysis

3.4. Data about the Use of Vitamin D supplements  

We changed the text to: Vitamin D Supplement Use by Participants 

  1. The following titles should be rewritten.

Table 1. Sample Characteristics, Demographics, and Categorical Variables.

We rewrote as: Sample Characteristics of the Study Participants

Table 2 Descriptive Statistics for Outcome Variables

 We rewrote as: Descriptive Statistics for Key Variables for Study Participants.

  1. What did FFR stand for in Table 3? We fixed this error, it should be FFQ
  2. There were some wrong spellings. We ran another thorough round and fixed all errors
